

# Tracking mosquito-borne diseases via social media: a machine learning approach to topic modelling and sentiment analysis

Song-Quan Ong[1,2] and Hamdan Ahmad[3]

[1] Institute of Tropical and Conservation, Universiti Malaysia Sabah, Kota Kinabalu, Sabah, Malaysia
[2] Department of Ecoscience and Arctic Research Centre, Aarhus University, Aarhus, Denmark
[3] Vector Control Research Unit, Universiti Sains Malaysia, Bayan Lepas, Penang, Malaysia

## ABSTRACT

Mosquito-borne diseases (MBDs) are a major threat worldwide, and public consultation on these diseases is critical to disease control decision-making. However, traditional public surveys are time-consuming and labor-intensive and do not allow for timely decision-making. Recent studies have explored text analytic approaches to elicit public comments from social media for public health. Therefore, this study aims to demonstrate a text analytics pipeline to identify the MBD topics that were discussed on Twitter and significantly influenced public opinion. A total of 25,000 tweets were retrieved from Twitter, topics were modelled using LDA and sentiment polarities were calculated using the VADER model. After data cleaning, we obtained a total of 6,243 tweets, which we were able to process with the feature selection algorithms. Boruta was used as a feature selection algorithm to determine the importance of topics to public opinion. The result was validated using multinomial logistic regression (MLR) performance and expert judgement. Important issues such as breeding sites, mosquito control, impact/funding, time of year, other diseases with similar symptoms, mosquito-human interaction and biomarkers for diagnosis were identified by both LDA and experts. The MLR result shows that the topics selected by LASSO perform significantly better than the other algorithms, and the experts further justify the topics in the discussion.

## INTRODUCTION

Mosquito-borne diseases (MBDs) are responsible for more than 400,000 deaths worldwide every year (*WHO, 2020*). In addition to general disease management and disease control, one of the pillars of MBD management is disease surveillance (*WHO, 2020*), where infected or potential patients and the vector mosquito are monitored in time and space. A surveillance project is usually initiated by the authority or government, which actively collects information on reported cases and surveillance of the vector mosquito by health personnel and health officials. To improve disease surveillance, many studies involve public

Corresponding author
Song-Quan Ong,
songguan26@gmail.com

comments and opinion surveys, in which citizens play a key role in providing information on infections, diseases, vector mosquito abundance and biodiversity, *etc.* (*Lin et al., 2016*; *Ong, Pauzi & Gan, 2022a*). Traditional methods for obtaining this information include home visits and manual surveys/interviews (*Safdar et al., 2016*), which are often time-consuming, costly, and labor-intensive (*Queirós, Faria & Almeida, 2017*). Data from social media offers an excellent alternative to capture public opinion. *Lim, Tucker & Kumara (2017)* demonstrated a bottom-up approach using an unsupervised machine learning model to identify symptoms of infectious diseases from social media data. *García-Díaz et al. (2018)*, on the other hand, developed a system for classifying tweets using supervised machine learning and compared the performance of three classifiers—J48, BayersNet and SMO—in classifying tweets about infectious diseases. Previous studies have also demonstrated the use of Latent Dirichlet Allocation (LDA) models and sentiment analysis to understand public opinion on a specific aspect of infectious diseases, *e.g.*, vaccination (*Jabalameli, Xu & Shetty, 2022*; *Xie et al., 2021*), spatial–temporal features (*Zhu et al., 2020*), detection (*Ye et al., 2016*). However, most of them focused on machine intelligence, with human experts hardly playing a role. For example, in mosquito-borne diseases, there are many concepts such as the name of the pathogen, symptoms in humans, breeding sites of mosquitoes, *etc.*

Machine intelligence tools primarily help with data aggregation, feature engineering and modelling. The involvement of human experts in data integrity is crucial to justify the methodology and outcome (*Bazoukis et al., 2022*). For example, the quality of the input data used to train a machine learning model must be checked for completeness, correctness, agreement, plausibility and timeliness through relatively simple automated approaches and targeted manual validation by a domain expert (*Verma et al., 2021*), otherwise the model would perform poorly in the real world. While the involvement of human experts is widely used in healthcare as an important approach to disease diagnosis and detection (*Bhandari & Reddiboina, 2019*; *Sevakula et al., 2020*; *Long & Ehrenfeld, 2020*), the field of mosquito-borne diseases illustrates the necessity and possibilities of involving human experts when analyzing social media data. *Long & Ehrenfeld (2020)* defined the role of human experts in preventing COVID-19 infections by obtaining real-time data, with epidemiologists and public health officials ensuring data integrity. *Bazoukis et al. (2022)* proposed a general framework and challenges for augmented intelligence and medical knowledge developers when integrating both machines and humans into a development pipeline. Of course, previous studies have attempted to incorporate the role of humans in the pipeline of systematic information retrieval, but many improvements are still needed to minimize bias and validate the results. Therefore, this study aims to demonstrate an advanced text analytics pipeline for mosquito-borne diseases based on text data from social media. More specifically, we aim to integrate human perception in the process of data cleaning, data annotation, topic selection and validation of a machine intelligence pipeline on topics related to mosquito-borne diseases on Twitter.

## MATERIALS & METHODS

### Text analytics framework

Figure 1 shows the pipeline of text analytics in this study and the roles of machine and human for each of the components. There are two platforms for the machine application: Rapidminer (version 9.10) for crawling tweets, topic and sentiment modelling and RStudio (version 4.1.1; *R Core Team, 2021*; *RStudio Team, 2021*) for the topic selection algorithm and validation using multinomial logistic regression. An expert was defined as a group of experts with direct responsibility or special interest in mosquito-borne diseases. This included representatives of researchers, government organizations and public health professionals. The academic qualification of the human experts was at least a master's degree and/or at least five years of experience in public health, epidemiology, medical entomology, and parasitology. A total of four human experts were involved, namely a scientific officer from the Ministry of Health, a senior lecturer from the Vector Control Research Unit, Universiti Sains Malaysia, a senior researcher from the Institute Tropical Biology and Conservation, Universiti Malaysia Sabah, and a senior researcher from Kasetsart University Thailand. The cross-validation was conducted by sharing the tweets/topics generated by Rapidminer on the cloud platform and allowing human experts to immediately comment on the relevance of the tweets/topics.

### Data set construction
#### *Collection, cleaning, and annotation of the data by a machine and a human expert*

Figure 2 shows the general workflow for creating the annotated dataset, which consists of latent Dirichlet allocation (LDA) for topic and sentiment analysis. We collected the social data by accessing the tweets from Twitter. Specifically, we used the Twitter search operator in Rapidminer to query the word "mosquito", "malaria", "dengue", "Zika" and/or "chikungunya" on Twitter from May to July 2022. The query and selection of keywords were based on the major vector-borne diseases published on *WHO (2020)*. For data cleaning, regular expressions were used to replace "http", "#tags" and "RT @" such as "http-https?://[-a-zA-Z0- 9+&@#/%? =~_|!:,.;]*[-a-zA-Z0-9+&@#/% =~_|]" and "RT @- RT \s*@[:]*:\s*[A-Za-z ]+" as well as the retweets. Experts then manually identified and removed the irrelevant and unrepresentative public tweets.

To identify the topics discussed on Twitter, we applied the latent Dirichlet assignment (LDA) algorithm (*Reyes-Menendez, Saura & Alvarez-Alonso, 2018*). This is an unsupervised machine learning method that allows the computation of words or documents in a corpus to be explained by latent groups, such as topics (*Aenishaenslin et al., 2013*; *Triantaphyllou, 2000*; *Ong, Pauzi & Gan, 2022a*). We used the RapidMiner operator "Extract Topic from Text" with 5,000 optimisation iterations. For sentiment annotation, we used the "Sentiment Extract" operator with the selection of Valence Aware Dictionary and sEntiment Reasoner (VADER) in RapidMiner to compute the sentiment score and annotate the tweets. VADER is a lexical and rule-based sentiment model (https://github.com/cjhutto/vaderSentiment) to score the text (*Hutto & Gilbert, 2014*). VADER is specifically tuned to the sentiments

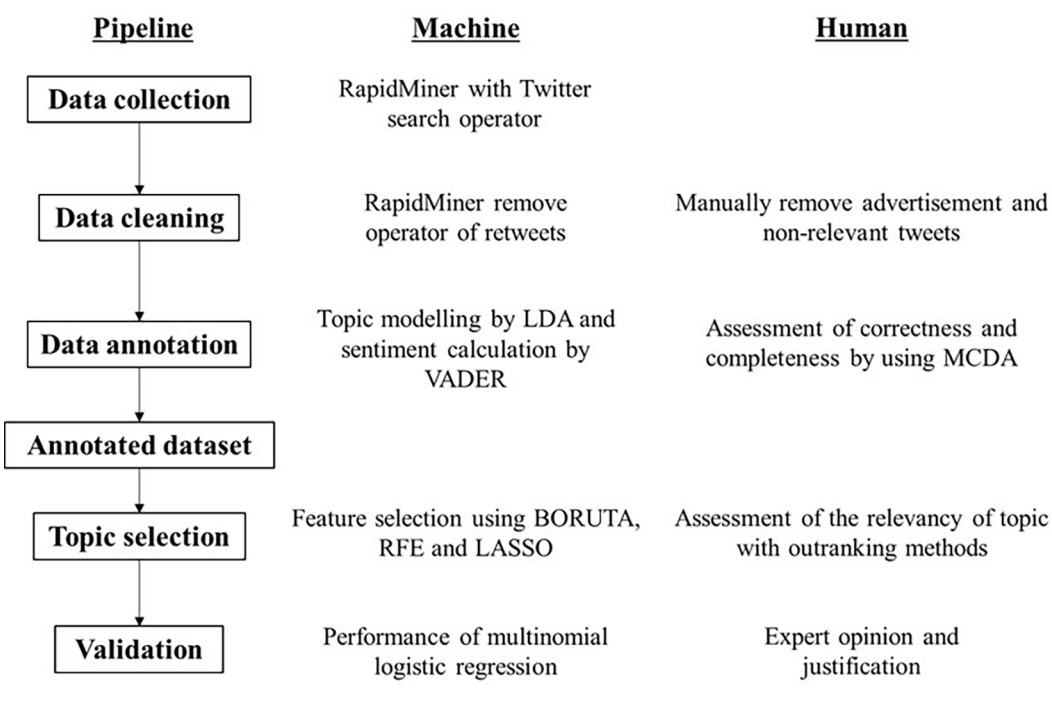

**Figure 1** Text analytics pipeline and the role of machine and human, respectively.

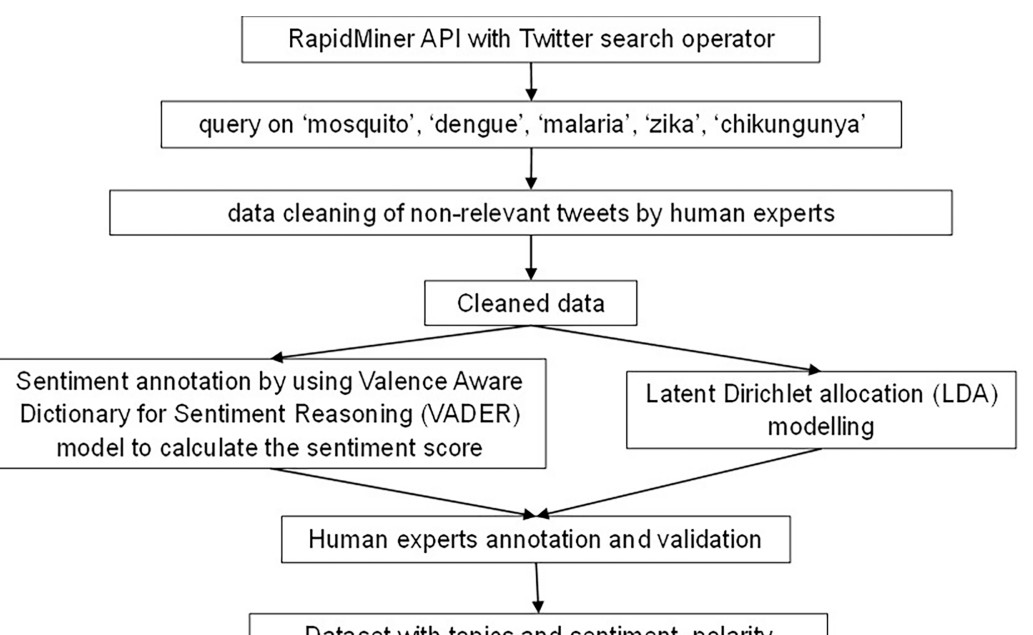

**Figure 2** Figure 2 shows the overall workflow to construct the annotated dataset.

expressed in social media and generates ratings based on a dictionary. This operator calculates the sum of all ratings of the sentiment words in the text and then outputs them.

To improve the result of the data annotation, we used the compensatory method, where a topic was proposed and selected based on a systematic evaluation (*Triantaphyllou, 2000*). For validation, each annotation of tweets (topic and sentiment) was rated independently by each expert. If the tweet contained information about a mosquito-borne disease, it was checked for topic relevance, otherwise it was filtered out. Following this process, a tweet was only selected if all experts agreed on its relevance to a mosquito-borne disease.

### Topic selection and validation based on public sentiments

One of the objectives of this study is to identify the issues that significantly influence public opinion. After experts annotated the topics generated from the LDA, the topics were processed using three feature selection algorithms—BORUTA, Recursive Feature Elimination (RFE) and Least Absolute Shrinkage and Selection Operator (LASSO)—to select the topics that significantly influence the polarity of sentiment. Experts assessed the relevance of the topics selected by the algorithm and made adjustments to the annotation. The result of the feature selection algorithms consisted of four groups of topics - All, BORUTA selected topics, RFE and LASSO. To validate the topic groups were able to predict sentiment polarity, we used a multinomial logistic regression that predicted three sentiments—positive, neutral and negative—using each of the four topic groups as input data. These topics served as independent variables or predictors, and polarity was the dependent variable. For example, the tweets with the topics of "symptoms", "mosquito-human interaction", "season", *etc.* were used as the data for the predictors to build a regression model to predict the polarity of the tweet. We built a total of four regression models (topics selected by BORUTA, RFE, LASSO and all topics) and randomly divided the data of tweets into the training set (80%) and the test set (20%). The logistic regression model was validated with a five-fold cross-validation. The evaluation matrices for accuracy, sensitivity and specificity were used to compare the performance of the models. To statistically compare the performance of the four regression models, a one-way ANOVA (SPSS 21.0; SPSS Inc., Chicago, IL, USA) was used to compare the scoring matrices at a $p = 0.05$ level.

## RESULTS

### Data set construction

After data cleaning, we obtained 6,263 tweets from a total of 25,000 tweets from Twitter. Figure 3 shows the distribution of tweets by keyword and the corresponding word cloud. Figure 3 shows that dengue had the highest number of tweets among mosquito-borne diseases and mosquito had the second highest number of tweets, which is consistent with the result of the word cloud for all diseases, which shows that "mosquito" has the highest TF-IDF. The search terms "Chikungunya" and "Zika" had a low number of tweets. Nevertheless, the names of other diseases such as "Ebola" and "Dengue" had a high weight, which could be due to the fact that the corresponding tweets contained multiple tags for

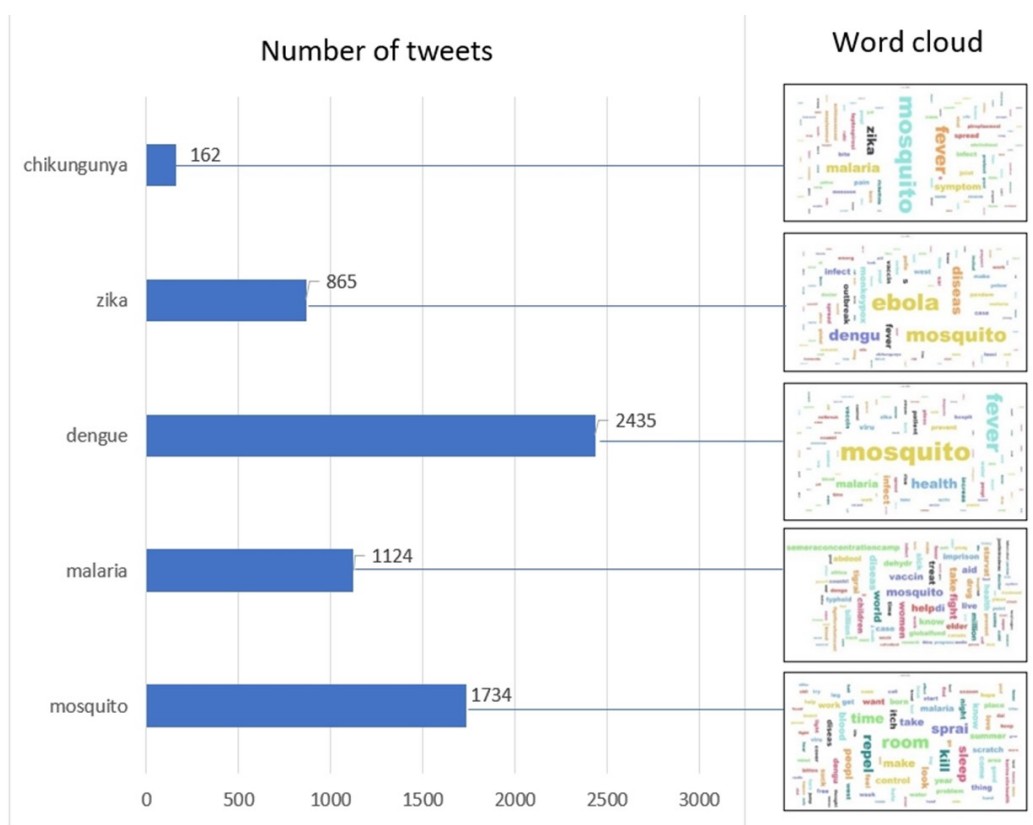

**Figure 3 Distribution of tweets by keyword and the corresponding word cloud.** The size of the word indicates the density of tweets mentioning the word, *e.g.*, the 2,435 tweets queried with "dengue" mainly mentioned "mosquito", "fever", "health" and "malaria", as indicated by the larger front size of the word in the word cloud.

more than one disease. The annotated data set has been uploaded and is publicly available in Figshare (https://doi.org/10.6084/m9.figshare.20493501.v1).

Based on the results of the expert annotation and LDA modelling, a total of 10 topics were identified in this study. Table 1 summarises the topics modelled with LDA, the example of words used for clustering in LDA modelling, and the experts' topic annotations. Meanwhile, sentiment polarity was calculated using the VADER model. After cross-validation by experts, Fig. 4 shows the overall distribution of polarity in this study. Based on the polarity result, we can see that the public perception of MBD is rather negative.

## Topic selection and validation

The topics selected by the feature selection algorithms are shown in Fig. 5. From the selection results, it can be seen that 'unsafe', 'pathogens' and "breeding sites" were not selected by any algorithm. In addition, five topics - "effort", "symptoms", "other

**Table 1  Summary of the topics and example of words that modelled by LDA and annotation by human expert.**

| Topic | Words used by LDA modelling | Annotation |
|---|---|---|
| 0 | back, million, help, fight, lives, save, end, COVID, GlobalFund | Efforts/funding to fight diseases |
| 1 | Died, dehydration, starvation, imprisoned | Symptoms/outcome of an outbreak |
| 2 | name, know, great, since, even, idea | Uncertain with something |
| 3 | Zika, virus, Ebola, COVID, flu, Covid, vaccine | Other diseases with similar symptoms |
| 4 | Mosquito, bites, fever | Mosquito-human interaction |
| 5 | control, health, fever, sleeping | Mosquito control |
| 6 | Water, breeding, mosquito, action, diseases, health, activities, government | Breeding places |
| 7 | Blood, hospital, platelets | Biomarkers for diagnosis |
| 8 | Cases, year, reported, DOH, July, June, country, January | Season |
| 9 | Virus, fever, Zika viruses, disease, infection | Pathogen |

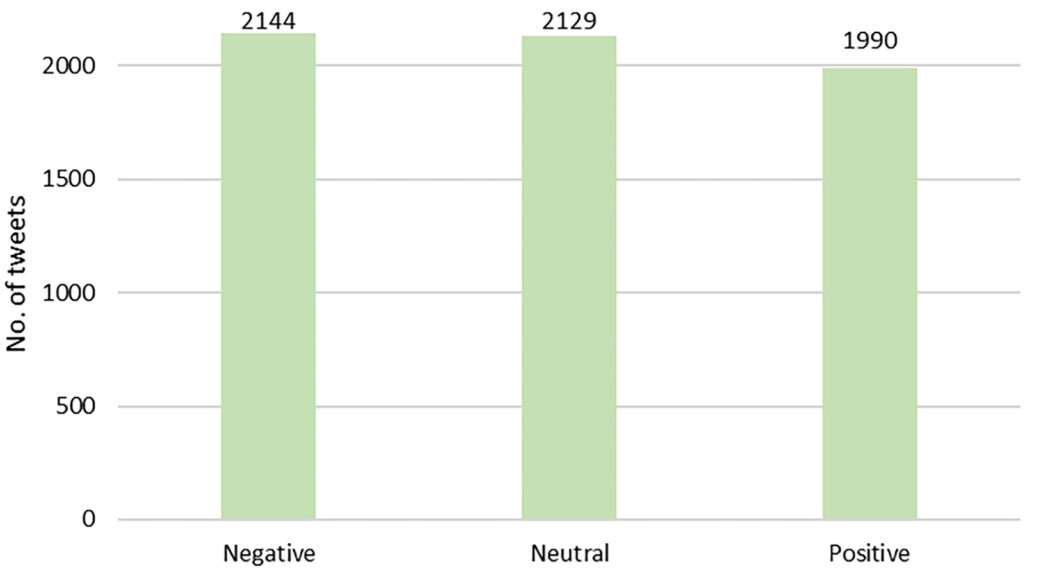

**Figure 4  Distribution of sentiment polarity of tweets in this study.**

diseases with similar symptoms'', "mosquito control" and "mosquito-human interaction" identified by BORUTA are common features used by other algorithms.

We used the predictive power of logistic regression to validate the topics and the predictive power of the regression, with the hypothesis that the performance of the model should be higher when the topics were more significant. Our result showed that the topics selected with LASSO performed significantly better at $p < 0.05$ than other feature groups, namely 'effort/funding', 'diseases with similar symptoms', 'mosquito-human interaction', 'symptoms', 'mosquito control', 'season', 'Biomarkers for diagnosis' Fig. 6 shows the performance (mean ± standard error as confidence interval) of the multinomial logistic regression using different feature groups of characteristics as independent variables to
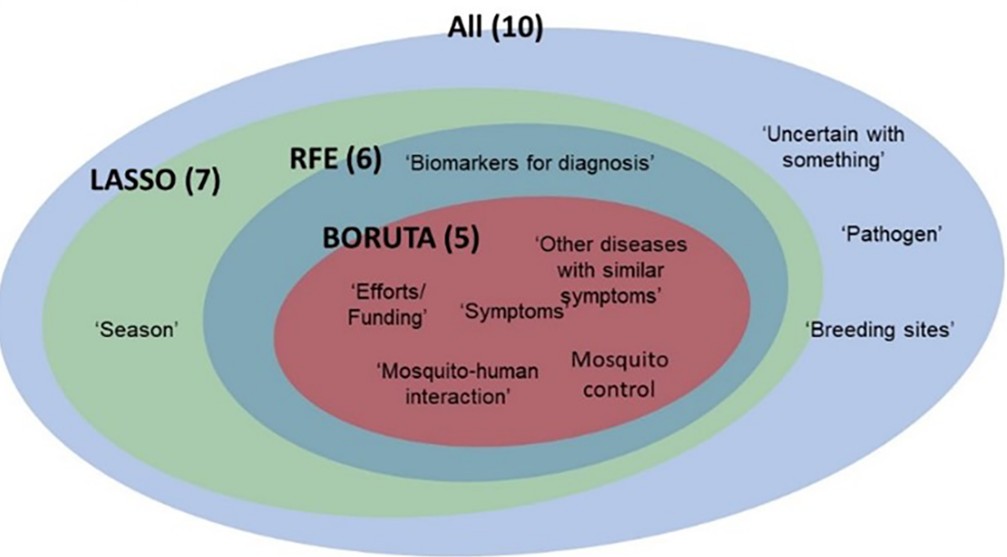

**Figure 5   Venn diagram of the topics selected by feature selection algorithm.**

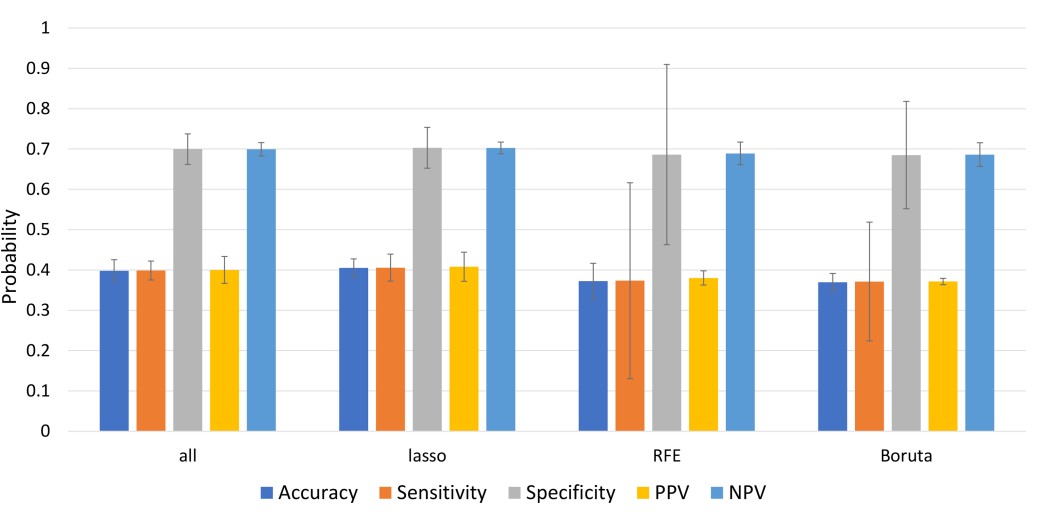

**Figure 6   Performance of multinomial logistics regression by using different selected topic group.** Mean ± standard error as the confidence interval.

predict the dependent variable: sentiment polarity. Table 2 shows the parameter estimates for each set of predictors used to predict sentiment polarity. Some of the predictors had a high significance value, suggesting that the variable could have a decisive influence on public sentiment. In the LASSO group, "season" was emphasised, which could make an important contribution to improving the performance of the prediction model. Some weak predictors such as "breeding site" and "pathogen" were not significant for all groups, which could be due to weak public education on the importance of these two topics.

**Table 2 Parameter estimation for each predictor variables.**

| Group of predictors that used for polarity prediction[a] | Variables | B | Standard Error | Sig. |
|---|---|---|---|---|
| ALL | Intercept | −87.016 | 86.344 | .012 |
| | Efforts/funding to fight diseases | 14.078 | 41.626 | .068 |
| | Symptoms/outcome of an outbreak | 23.045 | 61.832 | .011 |
| | Uncertain with something | −1.685 | 71.254 | .608 |
| | Other diseases with similar symptoms | 35.540 | 51.196 | .016 |
| | Mosquito-human interaction | 27.156 | 41.852 | .018 |
| | Mosquito control | 19.014 | 30.084 | .055 |
| | Breeding places | 2.074 | 23.080 | .691 |
| | Biomarkers for diagnosis | 10.745 | 11.025 | .076 |
| | Season | −22.087 | 51.080 | .005 |
| | Pathogen | 6.001 | 32.260 | .845 |
| BORUTA | Intercept | −109.401 | 91.128 | .009 |
| | Efforts/funding to fight diseases | 16.177 | 42.053 | .055 |
| | Symptoms/outcome of an outbreak | 25.901 | 21.126 | .009 |
| | Other diseases with similar symptoms | 38.604 | 41.852 | .012 |
| | Mosquito-human interaction | 28.066 | 50.024 | .008 |
| | Mosquito control | 20.118 | 70.074 | .063 |
| RFE | Intercept | −121.578 | 101.021 | .009 |
| | Efforts/funding to fight diseases | 13.178 | 22.053 | .036 |
| | Symptoms/outcome of an outbreak | 21.846 | 31.125 | .008 |
| | Other diseases with similar symptoms | −5.686 | 41.752 | .005 |
| | Mosquito-human interaction | 31.560 | 10.023 | .008 |
| | Mosquito control | 25.106 | 50.274 | .063 |
| | Biomarkers for diagnosis | 17.211 | 20.401 | .089 |
| LASSO | Intercept | −212.987 | 81.880 | .005 |
| | Efforts/funding to fight diseases | 12.054 | 51.224 | .016 |
| | Symptoms/outcome of an outbreak | 75.689 | 12.451 | .009 |
| | Other diseases with similar symptoms | 66.204 | 21.025 | .001 |
| | Mosquito-human interaction | 58.210 | 40.754 | .002 |
| | Mosquito control | 8.064 | 20.123 | .044 |
| | Biomarkers for diagnosis | 10.157 | 10.254 | .039 |
| | **Season** | 56.751 | 10.001 | .009 |

Notes.

[a] ALL, all topics used as predictors; BORUTA, the topics selected by BORUTA as predictors; LASSO, the topics selected by LASSO as predictors; RFE, the topics selected by RFE.
B is the logistic coefficient for each predictor variable for each alternative category of the outcome variable; sig., signficnat value at $p = 0.05$; $df = 6263$.

## DISCUSSION

According to the *WHO (2020)*, mosquito-borne diseases such as malaria, dengue, chikungunya and Zika are responsible for more than 700,000 deaths. As the diseases are transmitted by mosquito vectors, with the pathogen being transmitted by a specific mosquito (*e.g.*, the malaria parasite Plasmodium from an infected Anopheles mosquito), it was crucial for a surveillance program to understand the risk factors of the diseases in

the population. To obtain such information, public surveys were usually conducted. The surveys often asked about symptoms such as fever and rashes to understand the historical background of the participants. In addition, the surveys often asked about topics such as mosquito-human interaction, *e.g.*, previous encounters with insects or mosquito bites. However, conventional public surveys were labor-intensive and time-consuming, so text mining with machine learning could be a great help in obtaining this information. In contrast to the conventional machine learning feature selection pipeline, which mainly focuses on dimension reduction and feature importance to improve the performance of the prediction model, we emphasize the role of experts in an augmented text mining pipeline. As far as the authors were aware, the pipeline proposed in Fig. 1 is the first of its kind in augmented text analytics. The reason for augmenting the process is that the text used in mosquito-borne diseases can be specific and requires an extension of human experts' insights to obtain a more accurate annotation. For example, tweets from this study such as "DOH open to proposals of reusing anti-dengue vaccine, Dengvaxia, amid surge in cases. @MalayaNews" was commented as negative due to the key word "anti", but the introduction of a vaccine against dengue could be positive or neutral for disease management. Another example would be "Based on data from the Department of Health, the number of dengue cases in the Philippines has reached 51,622 from January to June this year, compared to 32,610 in the same period in 2021". was annotated as positive due to the keyword "up" but the content reflected a relatively negative message. Most of the tweets were adverts or irrelevant content, justifying the role of a human expert in cleaning the data, as the distinction between advert and public comment/post/message was difficult to make by the machine alone. The step of data understanding has been emphasized in many previous studies, (*e.g.*, *Berendt, Sammut & Webb, 2016*; *Hossain et al., 2021*; *Mukherjee & Sarkar, 2020*), in which performing text mining on news corpus and terminology from different domains was the main challenge in extracting information. Nevertheless, relatively few studies have proposed a clear framework that integrates expert knowledge into the machine intelligence pipeline.

One of the main problems with human experts is the potential bias caused by both humans and machines. This has been highlighted by the American Medical Association (AMA) (*Crigger & Khoury, 2019*). and in previous studies (*DeCamp & Lindvall, 2020*; *Livingston, 2020*; *Crigger et al., 2022*). which have proposed guidelines for the integration of human experts into artificial intelligence. In terms of human bias, this study has shown the tools that can be used to minimize the problem in most situations when the guidelines are applied, and multiple stakeholders agree on an opinion and judgement. The result is consistent with the findings of *Aenishaenslin et al. (2013)* and *Van Gennip, Hulshof & Lootsma (1997)* that a good level of agreement between stakeholders could usually be achieved, especially when commenting on and selecting topics. The tools were very useful in counteracting the cost and time required for a person to make decisions.

The topics selected with LASSO performed best in the multinomial logistic regression, suggesting that the topics significantly influence public sentiment ($p < 0.05$). This result is consistent with that of *Kouwayè, Fonton & Rossi (2015)*, who used LASSO to determine the risk factor of malaria, and that of *Ong, Ahmad & Mohd Ngesom (2021)*, who used

LASSO for real-time prediction of an outbreak of an endemic infectious disease. From the multinomial logistic regression result, three topics that were not selected by LASSO were 'Unsafe with something', 'Pathogens' and 'Breeding sites', suggesting that these topics are the least likely to influence public sentiment. For example, some of the tweets on 'Unsafe to handle' were: "Do home remedies work? Can papaya leaf juice help with dengue fever? What about neem leaves for chicken pox?", ".... The hospital has ruled out pneumonia, my only other concern is that it's dengue fever..".; "Did you know that MOSQUITOES DO PREFER TO BITE SOME PEOPLE OVER OTHERS?"; basically, users on social media were asking questions without saying much. Nevertheless, the content of the topic also suggests that there is a gap in the public's knowledge about the symptoms, treatments, types of pathogens and breeding sites of mosquito-borne diseases.

On the other hand, the RFE and LASSO algorithms highlight the importance of the issues 'biomarkers for diagnosis' and 'season'. If these topics significantly improve the performance of the multinomial logistic regression model, this could indicate some interesting information. Therefore, we further categorize the themes according to their polarity of sentiment so that experts can discuss them and extract further information. Figure 7 shows the distribution of polarity for each topic.

The topics 'Biomarkers for diagnosis', 'Efforts/funding' and 'Mosquito control' had a positive polarity. In relation to disease management, it was found that the public in this study perceived efforts, vector control and treatment positively. This finding was consistent with the reports of *Ong, Ahmad & Mohd Ngesom (2021)* and *Mashudi, Ahmad & Mohd Said (2022)* who reported that dengue cases were low to moderate during the duration of the pandemic and the public was more concerned about COVID-19 than mosquito-borne diseases, so the authorities' efforts were perceived more positively. The results were also agreed by *Moise et al. (2021)*, who indicated that mosquito control programs in Florida are vigilant and have considerable capabilities to control potential mosquito-borne diseases during the pandemic. On the other hand, the topic of 'season' is relatively neutral, as the majority of tweets contain recent news about dengue/disease cases in the state. For the topics 'Symptoms', 'other disease with similar symptoms' and 'mosquito-human interaction', the majority of the polarity was negative. For 'Symptoms', most tweets discussed the signs and symptoms or tragedy of a disease outbreak with many keywords such as 'angry', 'dying', 'sick', *etc.* The topic 'other diseases with similar symptoms' consists of many hashtags for non-mosquito-borne diseases such as Ebola, influenza and COVID19. The content of the tweets mainly dealt with the similarities of signs and symptoms (to confuse users) or efforts/funding/possible treatments. Mosquito-human interaction" is the reaction of humans after interacting with mosquitoes, *e.g.*, mosquito bites causing itching and their annoying effect causing discomfort in public.

This study has several limitations and challenges. First, the sample size obtained from social media is 25,000, and although this sample size is consistent with previous studies such as *Villagra et al. (2023)*, a larger sample size should be retrievable in the future. Secondly, the social media platform only consists of Twitter and excludes other platforms such as Facebook and Tik Tok which focus on sharing graphics and text. While the issue of open source for developers or data scientists is the main reason why Twitter was the mainstream
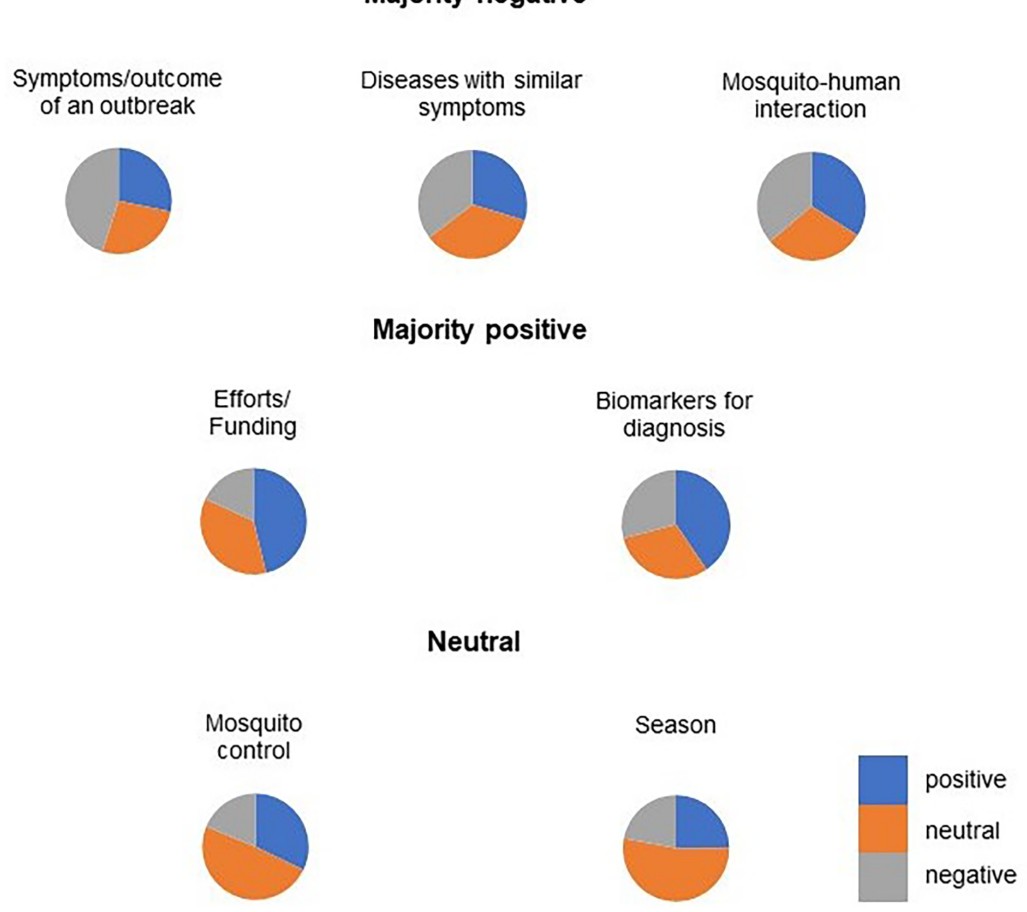

**Figure 7** **The distribution of polarity for the individual topic.** Three topics with a majority of negative polarity indicated by a larger grey area: "Symptom/outcome of an outbreak", "Diseases with similar symptoms". Two topics with a majority positive polarity, indicated by a larger blue area in the pie chart, were "Efforts/funding" and "Biomarkers for diagnosis". Two topics with neutral polarity, indicated by a larger orange area in the pie chart, were "Mosquito control" and "Season".

for text analytics, efforts may be made in the future to obtain permission to obtain the data from other social media. Third, as *Ong, Pauzi & Gan (2022b)* mention, data from social media comes with a lot of noise, bias, and ethical considerations. Future refinement of the text analytics pipeline with extended interaction with human experts could be key to the solution. Finally, information on the proportion of the population that has or does not have access to the internet was key to mosquito-borne diseases.

## CONCLUSIONS

We have shown the importance of human expert involvement in cleaning tweets (albeit with some automated cleaning operators/features), annotating topics/sentiments and validation. When applying text mining to the field of mosquito-borne diseases, the inclusion of human

experts in a text mining pipeline can help optimize the methodology and results due to the complexity of the pathogen, vector, and host components. Nevertheless, the analysis pipeline could be further improved by including more social media platforms and increasing the sample size so that a larger proportion of the community facing mosquito-borne diseases can be captured.

## ACKNOWLEDGEMENTS

The authors would like to thank the experts from the Ministry of Health Malaysia, Vector Control Research Unit, Universiti Sains Malaysia, Institute Tropical Biology and Conservation, Universiti Malaysia Sabah and Kasetsart University Thailand for their contributions in commenting and annotating the data.

### Funding
The authors received no funding for this work.

### Competing Interests
The authors declare there are no competing interests.

### Author Contributions
- Song-Quan Ong conceived and designed the experiments, performed the experiments, analyzed the data, prepared figures and/or tables, authored or reviewed drafts of the article, and approved the final draft.
- Hamdan Ahmad analyzed the data, authored or reviewed drafts of the article, and approved the final draft.

### Data Availability
The data is available at Figshare: Ong, Song-Quan (2022). Annotated dataset for augmented text mining study. figshare. Dataset. https://doi.org/10.6084/m9.figshare.20493501.v1.

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
