# Peer review of "Tracking mosquito-borne diseases via social media: a machine learning approach to topic modelling and sentiment analysis"

_PeerJ, doi:10.7717/peerj.17045_

## Round 0.1 · original submission · Major Revisions

The article is well written but based on the reviewer's comments needs Major revision, it would be better to fine-tune the methodology, figures, data presentation and breakdown, and address the lack of a conclusion.

Reviewer 1 has suggested that you cite specific references. You are welcome to add it/them if you believe they are relevant. However, you are not required to include these citations, and if you do not include them, this will not influence my decision

Reviewer 1 ·

Basic reporting

Thank you for giving me the possibility of reviewing this paper. I hope the authors find my comments productive and that it will help them to improve their research work.

In this article,Therefore, this study aims to demonstrate a text analytics pipeline to identify the topics from MBD that were discussed on Twitter and significantly inufluenced public opinion

Reinforce the sample size with the following reference
Villagra, N., Reyes-Menéndez, A., Clemente-Mediavilla, J., & Semova, D. J. (2023). Using algorithms to identify social activism and climate skepticism in user-generated content on Twitter. Profesional de la información, 32(3).

The reference format needs revision

Conclusions are poor. They need to be improved clarifying the improvement that this research has provided in the body of knowledge.

Experimental design

Justify keyword selection based on
WHO. Vector-borne diseases. who.int/news-room/fact-sheets/detail/vector-borne- diseases

explaining why you have based the selection on their research and how it fits your goal

And reinforce the methodology and topic for public health with these references
Reyes-Menendez, A., Saura, J. R., & Alvarez-Alonso, C. (2018). Understanding# WorldEnvironmentDay user opinions in Twitter: A topic-based sentiment analysis approach. International journal of environmental research and public health, 15(11), 2537.

Saura, J. R., Reyes-Menendez, A., & Thomas, S. B. (2020). Gaining a deeper understanding of nutrition using social networks and user-generated content. Internet Interventions, 20, 100312.

Validity of the findings

The analysis is appropriate and provides interesting results

Reviewer 2 ·

Basic reporting

The manuscript presents an innovative approach to tracking mosquito-borne diseases using social media data. The authors used machine learning techniques such as LDA and VADER to model the topics and calculate the sentiment polarities, respectively. They also used feature selection algorithms to determine the importance of the topics for public opinion and validated the result using multinomial logistic regression and expert judgement.

Overall, the manuscript is well written and clear. The introduction and background provide sufficient context and the references provided are relevant and up-to-date. However, there are some suggestions that could further enhance the study:

1. The proposed text analytics pipeline to discern MBD topics from Twitter that influence public opinion is innovative. However, it would enhance the manuscript's transparency if the raw data were made accessible, either through a public repository or a direct link, which is also required by the PeerJ policy.

2. Most figures are of good quality and well-labeled. It would be advisable to maintain consistency in font selection across all figures. The current font choice in Figure 7 appears somewhat unclear and deviates from the standard set by other figures. Opting for a clearer and more uniform font would enhance the figure's legibility and overall presentation.

3. Furthermore, certain figures, such as Figure 3, could benefit from further elaboration. While the representation is clear, diving deeper into the relationships between the keywords in Discussion section would enhance comprehension. Specifically, some readers might not be familiar with chikungunya, zika, dengue, and malaria within the field. A brief accompanying explanation would be beneficial.

Experimental design

1. A deeper dive into the rationale behind the chosen algorithm would be valuable. Comparing the results with alternative algorithms might underscore the efficacy of the selected approach. To bolster the study's conclusions, it would be advantageous to provide more granular data, such as exact performance metric values and statistical test p-values. Visual aids like tables or charts could further elucidate the findings.

2. Elaborating on the qualifications and recruitment process of the experts would enhance the credibility of their judgments. It would also be beneficial to understand the manual processes experts employed to determine topic relevancy.

Validity of the findings

1. While the manuscript mentions the use of various algorithms and methodologies, it would be beneficial to provide a more detailed breakdown of the underlying data. Ensuring that the data are robust, statistically sound, and controlled is paramount for the validity of the findings.

2. Given the focus on social media data, it would be valuable to discuss the generalizability of the findings. Considering a more varied dataset might ensure broader applicability of the results. Exploring other social media platforms like Facebook, Instagram, or Reddit could offer a more holistic view of public sentiment. How well do the results translate to broader populations or regions or platforms not covered in the study?

3. Addressing the study's limitations would provide a more rounded perspective. Discussing challenges inherent to using social media data, such as noise, bias, and ethical considerations, would be insightful. Suggestions for refining the text analytics pipeline, perhaps by integrating other machine learning techniques or data sources, would be a valuable addition.

·

Basic reporting

The article meets the journal’s guidelines.
The figures and tables in the manuscript have been checked. Overall, I commend the authors that the research layout is clear to understand. Professional English language has been used throughout. Figures are relevant and labelled. However, the code has not been provided which weakens the study.

Experimental design

no comment

Validity of the findings

Revisions
Perhaps, in line 28-29, “After data cleaning…”, the sentence needs to be higher up in the abstract because I assume the data cleaning was done before the feature selection process and all the other processes mentioned prior to this sentence. Since the study focused on the use of social media, it would help to shed a light on the internet services in the geographic location. For an instance, what proportions of population has access to affordable internet services which will probably impact the use of social media applications such as Twitter. Reporting the proportions will help set the ground work for the population of study. It is unclear which were the response variables and which were predictors for the regression models. Line 271: Rephrase sentence starting with “This concept …”.

---

## Round 0.2 · Minor Revisions

Reviewer 2 suggest Minor revision, thoroughly proof read article, and address experimental design in details.

Reviewer 2 ·

Basic reporting

The manuscript has made significant improvements in response to the feedback from the previous review cycle. However, there are still a few areas that could benefit from further clarification and investigation:

It is recommended to conduct a more thorough proofread of the manuscript. For instance, on Line 50, there is an issue with unintended underscore lines that need correction.

Experimental design

Providing additional details about the number of experts involved in the data assessment, as well as how they were engaged, will enhance the understanding of the study. It is also important to address how interpersonal variation was managed and how reproducibility was improved. Additionally, if cross-validation was performed between experts, please include that information for clarity."

Validity of the findings

N/A

·

Basic reporting

Authors have addressed the comments in my previous review of the article.
The article needs to further examination and is ready to be published.

Experimental design

NA

Validity of the findings

NA

Additional comments

NA

---

## Round 0.3 · accepted · Accept

The authors have addressed the Reviewer's comments.